

# Clinical characteristics and prognosis differences between isolated right and left ventricular myocardial infarction in the Chinese population: a retrospective study

Yuanyuan Chen[1,*], Chunyan Weng[1,*], Junru Wu[1], Yan Tang[1], Zhengxin Li[1], Qing Wen[1], Xuejing Sun[1], Mingxing Wu[2], Zhiliu Peng[2], Xiying Luo[3], Hong Yuan[4], Yao Lu[4], Wei Huang[1] and Jingjing Cai[1,4]

[1] Department of Cardiology, The Third Xiangya Hospital, Central South University, Changsha, Hunan, China
[2] Department of Cardiology, Xiangtan Central Hospital, Xiangtan, Hunan, China
[3] Department of Cardiology, The First Affiliated Hospital, University of South China, Hengyang, Hunan, China
[4] The Center of Clinical Pharmacology, The Third Xiangya Hospital, Central South University, Changsha, Hunan, China
* These authors contributed equally to this work.

Corresponding authors
Wei Huang,
huangderun2001@163.com
Jingjing Cai,
caijingjing83@hotmail.com

## ABSTRACT

**Background and aims:** Acute myocardial infarction (AMI) is divided into left ventricular myocardial infarction (LVMI) and right ventricular myocardial infarction (RVMI) according to the regions of myocardial ischemic necrosis. Clinical characteristics, treatment strategies, and prognosis differences between isolated RVMI and LVMI have not been well characterized. This study aimed to explore this difference of patients with isolated RVMI and LVMI.

**Methods:** This retrospective cohort study included 3,506 patients hospitalized with coronary angiography diagnosed type 1 myocardial infarction (MI). Characteristics of admission and treatment strategies were compared in patients with isolated RVMI and LVMI. COX proportional hazards models with and without inverse probability of treatment weighting (IPTW) adjustment were performed to estimate the difference in all-cause and cardiovascular mortality between the two groups.

**Results:** In this retrospective study, we found the frequency of isolated RVMI was significantly lower in the population than that of isolated LVMI (406 (11.6%) *vs* 3,100 (88.4%)). Patients with isolated RVMI have similar age, sex, and comorbidities to the patients with isolated LVMI. However, patients with isolated RVMI have lower heart rate and blood pressure, but higher rates of cardiogenic shock and atrioventricular block. It is noteworthy that patients with isolated RVMI are more likely to be complicated with the multivessel lesion. Patients with isolated RVMI have lower risk of all-cause mortality (HR 0.36; 95% CI [0.24–0.54], $p < 0.001$) and cardiovascular mortality (HR 0.37; 95% CI [0.22–0.62], $p < 0.001$) than patients with isolated LVMI.

**Conclusions:** This study showed that patients with isolated RVMI and LVMI have similar baseline characteristics. However, the clinical manifestations were different in the isolated RVMI and LVMI patients. This study revealed a better prognosis of

isolated RVMI patients compared to isolated LVMI, which indicates the ischemic region could be considered in AMI risk stratification models for better assessment of risk for adverse clinical events.

## KEY MESSAGES

- A total of 406 (11.6%) patients had isolated right ventricular myocardial infarction (RVMI) and 3,100 (88.4%) had left ventricular myocardial infarction (LVMI) in this study.
- Patients with isolated RVMI were more likely to have lower heart rate and blood pressure and suffer from cardiogenic shock and atrioventricular block on admission.
- Patients with isolated RVMI have lower risk of all-cause mortality and cardiovascular mortality during the 5.7-year follow-up than patients with isolated LVMI.

## INTRODUCTION

Despite an increase in the awareness and application of evidence-based treatment modalities for acute myocardial infarction (AMI) and its risk factors, AMI remains one of the leading causes of death worldwide (*Reed, Rossi & Cannon, 2017*). Patients with AMI are able to be stratified into left ventricular myocardial infarction (LVMI) and right ventricular myocardial infarction (RVMI) according to the regions of myocardial ischemic necrosis. Isolated RVMI frequently refers to infarcts in the blood supply area resulting from occlusion of the right coronary artery (RCA) proximal to the marginal branch (or the left circumflex artery in the left coronary dominance population) (*Bowers et al., 2002*; *Namana, Balasubramanian & Hollander, 2016*). Isolated LVMI includes infarcts in anterior, inferior, posterior, lateral and septal wall of the left ventricle. Due to the physical difference between the left-side and right-side heart, the presentation of RVMI have features, such as atrioventricular block, hypotension and cardiogenic shock. Such presentations changed the therapeutic regimen in RVMI patients.

Clinical features have been elaborated in a number of AMI subcategories, such as type 1 and type 2 MI, or patients with different demographic traits, such as age, sex, ethnicity, and pre-existing conditions (*Zhao et al., 2018*; *Ariss et al., 2021*; *Jangaard et al., 2017*; *Jacoby & Nesto, 1992*; *Dreyer et al., 2017*; *Krittanawong et al., 2021*). However, the difference in potential risk factors, clinical characteristics, in-hospital treatments, and prognosis between coronary angiography-diagnosed isolated RVMI and LVMI have not been well characterized. We, therefore, aimed to explore the difference in clinical characteristics, treatment strategies and prognosis in patients with isolated RVMI and LVMI in a retrospective cohort study. In addressing this hypothesis, we provide greater insight into better treatment and survival of isolated RVMI.

## METHODS

### Study design and patient population

This retrospective cohort study included 4,298 patients consecutive hospitalized with type 1 myocardial infarction (MI) in three hospitals in Hunan province from January 2013 to December 2017. There were 1,723 patients from the Third Xiangya Hospital of Central South University, 1,365 patients from the Xiangtan Central Hospital, and 1,210 patients from the First Affiliated Hospital of the University of South China. Patients with malignancy, severe and active autoimmune disease, trauma causing hemodynamic instability, active infection, severe renal impairment (estimated glomerular filtration rate (eGFR) <15 mL/min), and severe hepatic impairment (alanine transaminase (ALT) >400 U/L or total bilirubin (Tbil) >340 μmol/L) and whose age ≤18 years or ≥90 years and hospital stay less than 1 day were excluded. Patients were also excluded if the final diagnosis of RVMI or LVMI was unclear according to both coronary angiography and electrocardiogram and if the regions of infarction involved both the inferior or posterior wall of the left ventricle and right ventricular. There were 3,506 patients with type 1 MI in the final analysis (Fig. S1). The study was performed according to the 1975 Declaration of Helsinki and obtained the informed consent waiver that approved by the Third Xiangya Hospital of Central South University (NO: 2019-S489), the centre ethics committees of this study. This study was carried out according to the Strengthening the Reporting of Observational Studies in Epidemiology (STROBE) reporting guidelines.

### Clinical assessment

Clinical assessment of patients, including characteristics on admission and examination and treatment in hospital. Baseline characteristics on admission including (age, sex and health insurance (none, urban employee basic medical insurance, urban resident basic medical insurance, new rural cooperative medical scheme and others)), smoking status, body mass index (BMI), history of disease (diabetes, hypertension, stroke, angina, chronic heart failure, COPD (chronic obstructive pulmonary disease) and asthma), major presenting symptoms (chest pain, chest stuffiness, shortness of breath and syncope), cardiogenic shock on admission, laboratory test on admission (heart rate (HR), systolic blood pressure (SBP), diastolic blood pressure (DBP), hemoglobin (Hb), white blood cell (WBC), platelets count (PLT), peak cTnI, fasting blood glucose (FBG), alanine transaminase (ALT), low-density lipoprotein cholesterol (LDL-C), creatinine (Cr), serum potassium), and Killip classification indicating the severity of acute myocardial infarction. Findings and in-hospital procedures, including information on electrocardiography (ECG), echocardiography, coronary angiography, medicine treatment (angiotensin-converting enzyme inhibitors (ACEIs) or angiotensin receptor blockers (ARBs), beta-blockers, calcium channel blockers, statins, aspirin, clopidogrel, ticagrelor, intravenous nitrates and inotropes) and coronary revascularization.

### Follow-up and clinical outcomes

Information about clinical outcomes, including all-cause and cardiovascular mortality, were obtained from the Chinese Center for Disease Control and Prevention (CDC),

followed up from the time of hospitalization with AMI to 31 December 2021 or until death. The maximum follow-up time was 9.0 years (median and interquartile range (IQR): 5.7, 4.7–6.9 years). All-cause mortality meant death from any cause, and cardiovascular death were composed of cardiac death from myocardial infarction, hypertension and its complications, heart failure and malignant arrhythmia.

## Statistical analysis

Continuous variables were expressed as medians and interquartile ranges (IQRs) and categorical variables as numbers and percentages. Baseline characteristics, findings, in-hospital procedures, and clinical outcomes between those with isolated RVMI and those with LVMI were compared by ANOVA for continuous variables and chi-squared tests or Fisher's exact tests for categorical variables. All-cause mortality and cardiovascular mortality according to isolated RVMI and LVMI were plotted in inverse probability of treatment weighting (IPTW)-adjusted Kaplan-Meier curves, and the log-rank test was used for comparison of survival between the two groups.

Isolated LVMI was regarded as the reference group. Univariate and multivariable Cox proportional hazards models were used to evaluate the association of isolated RVMI *vs* LVMI with all-cause mortality and cardiovascular mortality. The multivariable models were adjusted for age, sex, health insurance, smoking status, BMI, history of disease (diabetes, hypertension, stroke, chronic heart failure and COPD or asthma), cardiogenic shock, ST-segment elevation myocardial infarction (STEMI) or Non-ST-segment elevation myocardial infarction (NSTEMI), laboratory test on admission (HR, SBP, DBP, FBG, LDL-C), LVEF, Killip classification, coronary angiography indicating multivessel stenosis, in-hospital medicine treatment (ACEIs or ARBs, beta-blockers, CCBs, statins, aspirin, clopidogrel, ticagrelor, intravenous nitrates and inotropes) and coronary revascularization. The multivariable models were adjusted for known confounders based on clinical consideration and previous research. The details for each abovementioned covariable were presented in Table S1. Besides, Cox proportional hazards models with an inverse probability of treatment weighting (IPTW) using the propensity score (PS) were applied to balance the difference in baseline characteristics and in-hospital findings and procedures between isolated RVMI and LVMI. In the IPTW approach, the PS was evaluated by logistic regression models, adjusting the same variables as the multivariable Cox proportional hazards model. Then each patient was weighted according to PS. To further minimize the selection bias, double robust models were conducted based on IPTW adjusted models through adjusting variables whose standardized mean difference (SDM) ≥0.05 (Table S2).

Subgroup analysis stratified according to sex (male or female), age (<60 years or ≥60 years), BMI (<24 kg/m$^2$ or ≥24 kg/m$^2$), smoking status (yes or no), history of diabetes (yes or no) and history of hypertension (yes or no) were performed to characterize potential effect modifications. Multivariable and IPTW-adjusted Cox proportional hazards models were used in subgroup analysis.

In the sensitivity analysis, propensity score matching methods matched 394 patients in two groups, respectively, through the abovementioned variables with a 1:1 ratio and multivariable Cox proportional hazards model, mixed effects COX model and

IPTW-adjusted Cox proportional hazards models further adjusted major presenting symptoms (chest pain, chest stuffiness, shortness of breath and syncope), echocardiography indices (left ventricular (LV) and right ventricular (RV)), arrhythmia (atrial fibrillation, atrioventricular block, bundle branch block, and bradycardia), and other laboratory tests (Hb, WBC, peak cTnI, ALT, PLT, Cr and serum potassium) were used to validate the stability of the results.

Model covariates with missing data were imputed before entering models using random forest with 10 iterations. All hypothesis testing was two-sided, and a $p$-value <0.05 was considered to indicate statistical significance. All statistical analyses were performed in R version 4.2.0 (*The R Foundation, 2023*).

## RESULTS

### Baseline characteristics in patients with isolated RVMI and LVMI

From January 2013 to December 2017, 3,506 patients with type 1 AMI were eligible for final analysis. The median (IQR) age was 64 (55–72) years, and 944 patients (26.9%) were females. Among 3,506 AMI patients, 406 individuals (11.6%) were diagnosed with isolated RVMI and 3,100 individuals (88.4%) were diagnosed with isolated LVMI. Most baseline characteristics, such as age, sex, and comorbidities, were comparable between isolated RVMI and LVMI. Differences included a higher proportion of patients have cardiogenic shock (11.6% *vs* 2.8%, $p < 0.001$) and atrioventricular blocker (10.3% *vs* 4.5%, $p < 0.001$) on admission in isolated RVMI group. In term of major presenting symptoms, patients with isolated RVMI tended to represent chest stuffiness (34.2% *vs* 28.8%, $p = 0.029$) and syncope (4.4% *vs* 1.5%, $p < 0.001$), while patients with isolated LVMI were more likely to had chest pain (79.1% *vs* 73.6%, $p = 0.015$). It is noteworthy that the median (IQR) heart rate (73 (62–85) beats/min *vs* 78 (69–89) beats/min, $p < 0.001$), systolic blood pressure (124 (110–140) mmHg *vs* 130 (116–146) mmHg, $p < 0.001$), diastolic blood pressure (77 (66–84) mmHg *vs* 80 (70–90) mmHg, $p < 0.001$) were significantly lower in patients with isolated RVMI than those in LVMI. The peak cTnI levels (1.72 [0.30–6.38] μg/L *vs* 1.19 [0.25–4.57] μg/L, $p = 0.007$) were higher in the isolated LVMI group than isolated RVMI group (Table 1).

### In-hospital medications and procedures in patients with isolated RVMI and LVMI

As the patients with isolated RVMI present lower heart rate and blood pressure, these patients have a lower rate of receiving antihypertensive drugs and vasodilator agents, such as ACEIs or ARBs (65.5% *vs* 76.3%, $p < 0.001$), beta-blockers (62.3% *vs* 74.9%, $p < 0.001$), intravenous nitrates (61.3% *vs* 69.0%, $p = 0.002$), than patients with isolated LVMI. In comparison, patients with isolated RVMI have a higher rate of inotropes treatment (31.1% *vs* 18.4%, $p < 0.001$) than isolated LVMI patients. Compared to isolated LVMI patients, there is a higher proportion of patients with NSTEMI than patients with isolated RVMI (46.6% *vs* 30.6%, $p < 0.001$). In addition, patients in isolated RVMI group were more likely to complicate with atrioventricular block (10.3% *vs* 4.5%, $p < 0.001$) and bradycardia (20.0% *vs* 6.7%, $p < 0.001$). Coronary angiography suggested that multivessel coronary

**Table 1 Baseline characteristics on admission in patients with right and left ventricular myocardial infarction.**

| Variables | All patients (*n* = 3,506) | Left ventricular myocardial infarction (*n* = 3,100) | Right ventricular myocardial infarction (*n* = 406) | *p*-value | SMD |
|---|---|---|---|---|---|
| Age, year (median (IQR)) | 64 (55–72) | 64 (55–72) | 65 (57–73) | 0.270 | 0.057 |
| Sex, *n* (%) | | | | 0.243 | 0.066 |
| Female | 944 (26.9) | 845 (27.3) | 99 (24.4) | | |
| Male | 2,562 (73.1) | 2,255 (72.7) | 307 (75.6) | | |
| BMI, kg/m² (median (IQR)) | 23.53 (22.67–24.84) | 23.56 (22.67–24.83) | 23.43 (22.71–24.94) | 0.822 | 0.020 |
| Smoke, *n* (%) | 1,684 (48.0) | 1,481 (47.8) | 203 (50.0) | 0.429 | 0.045 |
| Health insurance, *n* (%) | | | | 0.667 | 0.083 |
| None | 380 (10.8) | 337 (10.9) | 43 (10.6) | | |
| URBMI | 1,335 (38.1) | 1,171 (37.8) | 164 (40.4) | | |
| UEBMI | 640 (18.3) | 576 (18.6) | 64 (15.8) | | |
| NRCMS | 1,075 (30.7) | 948 (30.6) | 127 (31.3) | | |
| Others | 76 (2.2) | 68 (2.2) | 8 (2.0) | | |
| Killip classification, *n* (%) | | | | <0.001 | 0.218 |
| I | 2,373 (67.7) | 2,092 (67.5) | 281 (69.2) | | |
| II | 765 (21.8) | 696 (22.5) | 69 (17.0) | | |
| III | 203 (5.8) | 182 (5.9) | 21 (5.2) | | |
| IV | 165 (4.7) | 130 (4.2) | 35 (8.6) | | |
| Cardiac shock, *n* (%) | 135 (3.9) | 88 (2.8) | 47 (11.6) | <0.001 | 0.343 |
| Medical history, *n* (%) | | | | | |
| Diabetes | 683 (19.5) | 611 (19.7) | 72 (17.7) | 0.380 | 0.051 |
| Hypertension | 1,637 (46.7) | 1,448 (46.7) | 189 (46.6) | 0.994 | 0.003 |
| Angina | 126 (3.6) | 107 (3.5) | 19 (4.7) | 0.268 | 0.062 |
| Chronic heart failure | 75 (2.1) | 66 (2.1) | 9 (2.2) | 0.856 | 0.006 |
| COPD or asthma | 161 (4.6) | 137 (4.4) | 24 (5.9) | 0.221 | 0.067 |
| Stroke | 296 (8.4) | 268 (8.6) | 28 (6.9) | 0.273 | 0.065 |
| Cardiac arrest | 57 (1.6) | 51 (1.6) | 6 (1.5) | 0.966 | 0.013 |
| Presenting symptoms | | | | | |
| Chest pain | 2,750 (78.4) | 2,451 (79.1) | 299 (73.6) | 0.015 | 0.128 |
| Chest stuffiness | 1,033 (29.5) | 894 (28.8) | 139 (34.2) | 0.029 | 0.116 |
| Shortness of breath | 477 (13.6) | 424 (13.7) | 53 (13.1) | 0.789 | 0.018 |
| Syncope | 64 (1.8) | 46 (1.5) | 18 (4.4) | <0.001 | 0.175 |
| Vital parameters and laboratory findings on admission, median (IQR) | | | | | |
| HR, beats/min | 78 (68–88) | 78 (69–89) | 73 (62–85) | <0.001 | 0.282 |
| SBP, mmHg | 130 (115–145) | 130 (116–146) | 124 (110–140) | <0.001 | 0.215 |
| DBP, mmHg | 80 (70–90) | 80 (70–90) | 77 (66–84) | <0.001 | 0.259 |
| Hb, g/L | 129 (118–141) | 129 (118–142) | 126 (118–137) | 0.004 | 0.133 |
| WBC, 10⁹/L | 9.41 (7.26–11.66) | 9.44 (7.31–11.69) | 9.04 (7.08–11.51) | 0.095 | 0.063 |
| PLT, 10⁹/L | 200.00 (162.00–235.75) | 202.00 (162.75–236.25) | 189.00 (155.25–226.00) | 0.003 | 0.107 |

| Table 1 (continued) | | | | | |
|---|---|---|---|---|---|
| Variables | All patients (n = 3,506) | Left ventricular myocardial infarction (n = 3,100) | Right ventricular myocardial infarction (n = 406) | p-value | SMD |
| Peak cTnI, µg/L | 1.64 (0.29–6.07) | 1.72 (0.30–6.38) | 1.19 (0.25–4.57) | 0.007 | 0.107 |
| FBG, mmol/L | 6.44 (5.22–8.40) | 6.46 (5.22–8.40) | 6.30 (5.17–8.39) | 0.139 | 0.085 |
| ALT, U/L | 34.00 (20.60–48.00) | 34.10 (20.60–48.00) | 32.30 (20.42–46.85) | 0.403 | 0.021 |
| LDL-C, mmol/L | 2.65 (2.09–3.13) | 2.66 (2.10–3.13) | 2.60 (2.04–3.05) | 0.242 | 0.032 |
| Cr, µmol/L | 83.35 (70.00–96.00) | 83.00 (69.80–96.00) | 87.00 (71.08–100.00) | 0.021 | 0.057 |
| Serum potassium, mmol/L | 3.98 (3.70–4.27) | 3.98 (3.70–4.26) | 4.00 (3.68–4.31) | 0.373 | 0.051 |

**Note:**

SMD, standard mean difference; IQR, interquartile range; BMI, body mass index; UEBMI, urban employee basic medical insurance; URBMI, urban resident basic medical insurance; NRCMS, new rural cooperative medical scheme; COPD, chronic obstructive pulmonary disease; HR, heart rate; SBP, systolic blood pressure; DBP, diastolic blood pressure; Hb, hemoglobin; WBC, white blood cell; PLT, platelets count; FBG, fast blood glucose; ALT, alanine transaminase; LDL-C, serum low density lipoprotein-cholesterol; Cr, creatinine.

lesion was more common in patients with isolated RVMI (23.9% vs 14.7%, $p < 0.001$) and these patients have a higher rate of coronary revascularization (73.2% vs 67.5%, $p = 0.025$). Impairment of left ventricular ejection fractions (LVEF) was more severe in patients with isolated LVMI than that in patients with isolated RVMI (52 (44–60)% vs 56 (50–61)%, $p < 0.001$) (Table 2).

## Long-term all-cause and cardiovascular mortality in patients with isolated RVMI and LVMI

The median (IQR) length of hospital stays of patients with isolated RVMI and LVMI were similar (9 (7–12) days vs 9 (7–12) days, $p = 0.064$). The median (IQR) duration of follow-up was 5.7 (4.7–6.9) years (Table 2). There were 37 all-cause deaths (9.1%) in the isolated RVMI and 680 (21.9%) all-cause deaths in the isolated LVMI. In the isolated RVMI group, a total of 21 (5.2%) patients died of cardiovascular disease, while a total of 427 (12.8%) patients died of cardiovascular in the isolated LVMI group (Table 3). After IPTW adjustment for age, sex, health insurance, smoking status, BMI, history of disease (diabetes, hypertension, stroke, chronic heart failure and COPD or asthma), cardiogenic shock, STEMI or NSTEMI, laboratory test on admission (HR, SBP, DBP, FBG, LDL-C), Killip classification, coronary angiography indicating multivessel stenosis, in-hospital medicine treatment (ACEIs or ARBs, beta-blockers, statins, aspirin, clopidogrel, ticagrelor, intravenous nitrates, and inotropes) and coronary revascularization, the multivariable Cox models revealed that patients suffered isolated RVMI has a lower risk of all-cause mortality (HR 0.36; 95% CI [0.24–0.54], $p < 0.001$) and cardiovascular mortality (HR 0.37; 95% CI [0.22–0.62], $p < 0.001$) compared to patients with isolated LVMI (Table 4; Fig. 1).

## Subgroup analysis and sensitivity analysis

We further compared the all-cause and cardiovascular mortality between isolated RVMI and LVMI in subgroups with sex, age, BMI, smoking status, diabetes and hypertension. The all-cause mortality and cardiovascular mortality were lower in the subgroups of

**Table 2 In-hospital treatment and procedure in patients with right and left ventricular myocardial infarction.**

| Variables | All patients ($n$ = 3,506) | Left ventricular myocardial infarction ($n$ = 3,100) | Right ventricular myocardial infarction ($n$ = 406) | $p$-value | SMD |
|---|---|---|---|---|---|
| In-hospital medications, $n$ (%) | | | | | |
| ACEIs or ARBs | 2,632 (75.1) | 2,366 (76.3) | 266 (65.5) | <0.001 | 0.240 |
| CCBs | 571 (16.3) | 499 (16.1) | 72 (17.7) | 0.442 | 0.044 |
| Beta-blockers | 2,575 (73.4) | 2,322 (74.9) | 253 (62.3) | <0.001 | 0.274 |
| Statins | 3,409 (97.2) | 3,011 (97.1) | 398 (98.0) | 0.379 | 0.059 |
| Aspirin | 3,376 (96.3) | 2,983 (96.2) | 393 (96.8) | 0.664 | 0.031 |
| Clopidogrel | 2,774 (79.1) | 2,435 (78.5) | 339 (83.5) | 0.025 | 0.126 |
| Ticagrelor | 531 (15.1) | 489 (15.8) | 42 (10.3) | 0.005 | 0.162 |
| Intravenous nitrates | 2,388 (68.1) | 2,139 (69.0) | 249 (61.3) | 0.002 | 0.162 |
| Inotropes | 698 (19.9) | 571 (18.4) | 127 (31.3) | <0.001 | 0.301 |
| Electrocardiograph, $n$ (%) | | | | | |
| STEMI | 2,369 (67.6) | 2,152 (69.4) | 217 (53.4) | <0.001 | 0.333 |
| NSTEMI | 1,137 (32.4) | 948 (30.6) | 189 (46.6) | | |
| Ventricular fibrillation | 6 (0.2) | 5 (0.2) | 1 (0.2) | 0.522 | 0.019 |
| Atrial fibrillation | 122 (3.5) | 106 (3.4) | 16 (3.9) | 0.693 | 0.028 |
| New-onset atrial fibrillation | 86 (2.5) | 75 (2.4) | 11 (2.7) | 0.854 | 0.018 |
| Atrioventricular block | 180 (5.1) | 138 (4.5) | 42 (10.3) | <0.001 | 0.227 |
| Bundle branch block | 235 (6.7) | 210 (6.8) | 25 (6.2) | 0.718 | 0.025 |
| Bradycardia | 289 (8.2) | 208 (6.7) | 81 (20.0) | <0.001 | 0.397 |
| Coronary angiography | | | | | |
| Coronary dominance, $n$ (%) | | | | 0.829 | 0.032 |
| Balanced | 1,725 (49.2) | 1,520 (49.0) | 205 (50.5) | | |
| Left coronary dominance | 232 (6.6) | 207 (6.7) | 25 (6.2) | | |
| Right coronary dominance | 1,549 (44.2) | 1,373 (44.3) | 176 (43.3) | | |
| Multivessel lesion (%) | 553 (15.8) | 456 (14.7) | 97 (23.9) | <0.001 | 0.234 |
| Echocardiography | | | | | |
| Aortic regurgitation, $n$ (%) | | | | 0.258 | 0.097 |
| Mild | 1,172 (33.4) | 1,052 (33.9) | 120 (29.6) | | |
| Moderate | 56 (1.6) | 50 (1.6) | 6 (1.5) | | |
| Severe | 6 (0.2) | 5 (0.2) | 1 (0.2) | | |
| Tricuspid regurgitation, $n$ (%) | | | | 0.136 | 0.109 |
| Mild | 1,716 (48.9) | 1,524 (49.2) | 192 (47.3) | | |
| Moderate | 249 (7.1) | 222 (7.2) | 27 (6.7) | | |
| Severe | 32 (0.9) | 24 (0.8) | 8 (2.0) | | |
| Pulmonic regurgitation, $n$ (%) | | | | 0.243 | 0.098 |
| Mild | 514 (14.7) | 442 (14.3) | 72 (17.7) | | |
| Moderate | 8 (0.2) | 7 (0.2) | 1 (0.2) | | |
| Severe | 1 (0.0) | 1 (0.0) | 0 (0.0) | | |
| Decrease left ventricular compliance, $n$ (%) | 1,735 (49.5) | 1,523 (49.1) | 212 (52.2) | 0.264 | 0.062 |
| LVEF, % (median (IQR)) | 53 (45–60) | 52 (44–60) | 56 (50–61) | <0.001 | 0.312 |

| Variables | All patients (*n* = 3,506) | Left ventricular myocardial infarction (*n* = 3,100) | Right ventricular myocardial infarction (*n* = 406) | *p*-value | SMD |
|---|---|---|---|---|---|
| LA, mm (median (IQR)) | 33 (30–36) | 33 (30–36) | 33 (30–36) | 0.588 | 0.034 |
| RA, mm (median (IQR)) | 32 (30–35) | 32 (30–35) | 32 (30–35) | 0.019 | 0.134 |
| LV, mm (median (IQR)) | 49 (45–52) | 49 (45–52) | 48.50 (45–50) | 0.031 | 0.124 |
| RV, mm (median (IQR)) | 27 (20–29) | 26 (20–29) | 27 (19–30) | 0.330 | 0.029 |
| Revascularization, n (%) | 2,390 (68.2) | 2,093 (67.5) | 297 (73.2) | 0.025 | 0.124 |
| Length of stay, day (median (IQR)) | 9 (7–12) | 9 (7–12) | 9 (7–12) | 0.064 | 0.088 |

**Note:**
SMD, standard mean difference; ACEIs or ARBs, angiotensin-converting enzyme inhibitors or angiotensin receptor blockers; CCBs, calcium channel blockers; STEMI, ST-segment elevation myocardial infarction; NSTEMI, Non-ST-segment elevation myocardial infarction; LVEF, left ventricular ejection fraction; IQR, inter quartile range; LA, left atrial; RA, right atrial; LV, left ventricular; RV, right ventricular.

**Table 3 Mortalities in patients with right and left ventricular infarction.**

| Variables | All patients (*n* = 3,506) | Left ventricular myocardial infarction (*n* = 3,100) | Right ventricular myocardial infarction (*n* = 406) | *p*-value | SMD |
|---|---|---|---|---|---|
| All-cause mortality (%) | 717 (20.5) | 680 (21.9) | 37 (9.1) | <0.001 | 0.360 |
| Cardiovascular mortality (%) | 448 (12.8) | 427 (13.8) | 21 (5.2) | <0.001 | 0.297 |

**Note:**
SMD, standard mean difference.

**Table 4 The risk of all-cause mortality and cardiovascular mortality in patients with right and left ventricular myocardial infarction.**

| Outcomes | Unadjusted HRs (95% CI) | *p*-value | Adjusted HRs[1] (95% CI) | *p*-value |
|---|---|---|---|---|
| All-cause mortality | 0.39 [0.28–0.54] | <0.001 | 0.34 [0.24–0.48] | <0.001 |
| Cardiovascular mortality | 0.35 [0.23–0.55] | <0.001 | 0.29 [0.18–0.46] | <0.001 |
|  | IPTW-adjusted HRs[2] (95% CI) | *p*-value | Double robust adjusted HRs[3] (95% CI) | *p*-value |
| All-cause mortality | 0.40 [0.25–0.62] | <0.001 | 0.36 [0.24–0.54] | <0.001 |
| Cardiovascular mortality | 0.43 [0.24–0.78] | 0.005 | 0.37 [0.22–0.62] | <0.001 |

**Notes:**
HR, hazard ratio; 95% CI, 95% confidence interval.
[1] Adjusted for age, sex, health insurance, BMI, smoking status, in-hospital medications (ACEIs or ARBs, beta-blockers, CCBs, statins, aspirin, clopidogrel, intravenous nitrates and inotropes), medical history (diabetes, hypertension, stroke, chronic heart failure and COPD or asthma), STEMI or NSTEMI, cardiogenic shock, multivessel lesion, vital parameters and laboratory findings on admission (HR, SBP, DBP, FBG and LDL-C), LVEF, Killip classification and status of revascularization.
[2] Adjusted for confounders mentioned above.
[3] The double robust model additionally adjusted confounders with SMD ≥0.05 after IPTW adjustment, which were age, health insurance, clopidogrel, Killip classification, STEMI or NSTEMI, HR, SBP, DBP, FBG, LDL-C, chronic heart failure and multivessel lesion and status of revascularization.

patients with isolated RVMI compared to the corresponding subgroups with isolated LVMI, except for patients with diabetes (Figs. S2 and S3). In patients with diabetes, all-cause mortality and cardiovascular mortality risks were similar in isolated RVMI and LVMI groups (Figs. S2 and S3). Obesity patients (BMI ≥ 24 kg/m$^2$) and female patients had no significant difference in cardiovascular mortality between isolated RVMI and LVMI (Fig. S3).

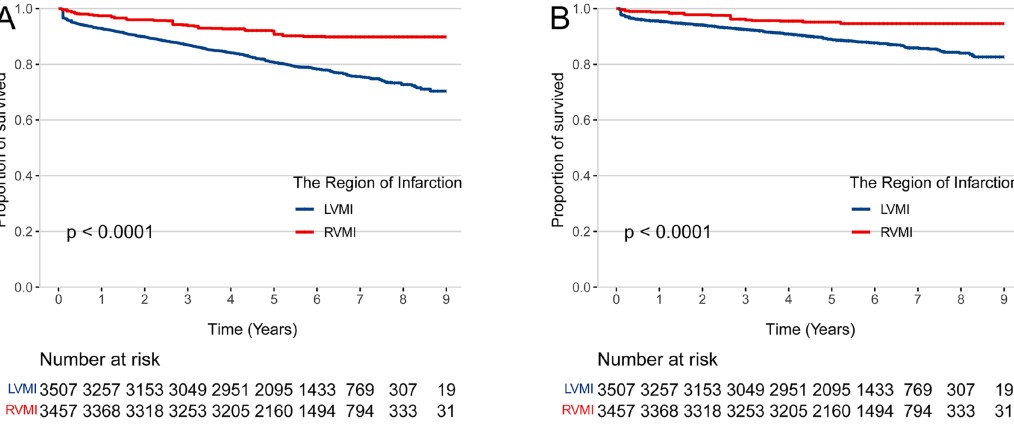

**Figure 1 The risk of mortality in patients with right and left ventricular myocardial infarction.** IPTW-adjusted Kaplan-Meier curves for all-cause mortality (A) and cardiovascular mortality (B) in patients hospitalized with right *vs* left ventricular myocardial infarction. RVMI, right ventricular myocardial infarction; LVMI, left ventricular myocardial infarction.

Four sensitivity analyses further confirmed the robustness of the association between the regions of myocardial infarction and prognosis. In Model 1, we used propensity score matching the abovementioned confounders in the Cox model. In Model 2, we adjusted additional confounders, including laboratory findings on admission (Hb, WBC, peak cTnI, ALT, PLT and serum potassium), major presenting symptoms (chest pain, chest stuffiness, shortness of breath and syncope), arrhythmia (atrial fibrillation, atrioventricular block, bundle branch block and bradycardia), echocardiography indices (LV and RV) besides abovementioned confounders. In Model 3, we used IPTW method to adjust for the same confounders as model 2. In Model 4, we adjusted confounders with SMD ≥ 0.05 after IPTW adjustment in model 3, which were age, health insurance, Hb, WBC, peak cTnI, PLT, serum potassium, LDL-C, chest stuffiness, STEMI or NSTEMI, bundle branch block, bradycardia and multivessel lesion. In Model 5, we performed the mixed effects COX model using the three hospitals as a random effect. The associations between the regions of myocardial infarction and clinical outcomes remained consistent and statistically significant after sensitive analyses (Table S3).

## DISCUSSION

In this retrospective study, we found the frequency of isolated RVMI was remarkably lower in the population than that of isolated LVMI. Patients with isolated RVMI have similar age, sex and comorbidities to those with isolated LVMI. Due to the physical difference between left-side and right-side heart, the patients with isolated RVMI have lower heart rate and blood pressure and higher rates of cardiogenic shock and atrioventricular block. Notably, patients with isolated RVMI are more likely to be complicated with the multivessel lesion. However, compared to isolated LVMI, patients with isolated RVMI have a lower risk of all-cause mortality and cardiovascular mortality during the 5.7-year follow-up.

Despite an increased prevalence of immediate revascularization and medication, the incidence of complications and mortality aftermath of AMI remain high globally (*Zhou et al., 2019*; *Roth et al., 2020*). Of note, RVMI is complicated with approximately 30% to 50% of cases of inferior-posterior wall myocardial infarction due to acute occlusion of the right coronary artery (RCA) (*Haji & Movahed, 2000*; *Shah et al., 1985*; *Cabin et al., 1987*; *Legrand et al., 1983*), while isolated RVMI considered to be rare (*Turkoglu, Erden & Ozdemir, 2008*; *Andersen, Falk & Nielsen, 1987*). Therefore, many previous studies focused on the prognostic impact of LVMI combined with RVMI (*Assali et al., 2007*; *Liao et al., 2020*; *Stiermaier et al., 2020*; *Hu et al., 2022*). This is the first cohort study to evaluate outcomes in patients with isolated RVMI *vs* LVMI. Our results suggested, even after adjusting for potential confounders, patients with RVMI have better prognostic in all-cause mortality and cardiovascular mortality than patients with isolated LVMI. The possible explanations for the difference could be as follow: first, a thin-walled structure of the right ventricular wall, which has lower wall pressure and lower demand for oxygen (*Jeffers, Boyd & Parks, 2022*); second, during systolic and diastolic perfusion of the right ventricle, hemodynamic stress improves the ability to take up oxygen, providing greater oxygen delivery to the RCA blooding supply region (*Crystal, Silver & Salem, 2013*); third, the right ventricular has rich collateral circulation stem from the left coronary artery and has direct blood supply from the right ventricular cavity by the thebesian veins (*Farrer-Brown, 1968*). In addition, this study also found that patients with RVMI were prone to cardiogenic shock on admission, which is consistent with previous findings (*Pfisterer, 2003*; *Jain et al., 2021*). The development of cardiogenic shock depends on the degree of right ventricular free wall dysfunction and the presence of right atrial ischemia and left ventricular injury (*Femia et al., 2021*). In patients with RVMI, acute proximal occlusion of the right ventricular marginal artery could impair perfusion of the ventricular free wall and contractile function of the right ventricular, leading to diminished transpulmonary delivery of left ventricular preload and resulting in low-output hypotension and cardiogenic shock (*Kakouros & Cokkinos, 2010*).

As the patients with RVMI present lower heart rate and blood pressure, these patients have a lower rate of receiving vasodilator agents and antihypertensive drugs. In a small case-control study of 40 patients with inferior wall myocardial infarction, hypotension developed in half of the patients treated with nitrates on admission, including 15 patients with RVMI, compared with only two patients without hypotension. However, it has been suggested that taking nitrates in patients with RVMI and inferior wall MI did not significantly increase the risk of adverse events (including hypotension, cardiogenic shock, cardiac arrest and death). In another study involving 82 patients receiving nitrates, AEs occurred in 11 (57.9%) of patients with inferior myocardial infarction complicated with RVMI and were found in 10 patients with inferior myocardial infarction without ($\Delta$20.9%, 95% CI [−7.8% to 45.4%], $p = 0.200$) (*McConnell et al., 2017*). In general, according to the ACCF/AHA Guidelines for STEMI management, patients with RVMI, hypotension, or bradycardia are not required to receive nitrates therapy (*O'Gara et al., 2013*). Antihypertensive reagents routinely used in LVMI patients, such as ACEIs and beta-blockers, should be applied carefully in patients who suffered RVMI due to reducing RV preload and

causing catastrophic haemodynamic compromise (*Kakouros & Cokkinos, 2010*; *Goldstein, 2012*). Therefore, antihypertensive drugs and vasodilator agents should be used cautiously in patients with RVMI (*Ferguson et al., 1989*). Similarly, the use of diuretics in patients with RVMI carries risks of exacerbating preload, it should be used cautiously (*Femia et al., 2021*).

Patients with RVMI should be optimized for right ventricular preload with inotropic reagents and intravenous fluids in addition to the cautious taking of antihypertensive drugs, nitrates and diuretics. An experiment in dog models of RVMI showed that the aortic pressure and cardiac output increased by 35% (*Goldstein et al., 1983*). In addition, although no animal studies or randomized trials have investigated the efficiency of inotropes, there is some evidence that dobutamine could improve cardiac index and cardiac output in patients with RVMI (*Ho & Nihoyannopoulos, 2006*). In another research involving volume loading *vs* dobutamine, 10 of 13 patients with RVMI first received a normal saline infusion in 200–800 ml, but cardiac index has not been improved significantly (from $1.9 \pm 0.5$ to $2.1 \pm 0.4$ l/min/m$^2$). Nevertheless, infusion of dobutamine after appropriate intravenous fluid infusion enabled a significantly improve cardiac index (from $2.0 \pm 0.4$ to $2.7 \pm 0.5$ l/min/m$^2$) (*Dell'Italia et al., 1985*). This might be explained by volume loading only improves pulmonary arterial wedge pressure instead of true preload or end-diastolic volume of the left ventricular to well meet the request for cardiac output, but dobutamine infusion could enhance ventricular myocardial contractility and directly improve cardiac index (*Dell'Italia et al., 1985*).

Ultimately, right ventricular output may be further reduced by an increase in afterload, so treatment to improve afterload should not be ignored. An animal model of RVMI showed that levosimendan, one of the inotropic reagents, not only enhanced myocardial contractility and improved haemodynamics, but slightly reduced afterload (*Missant et al., 2007*). In a recent meta-analysis of eight original studies involving 390 participants with right ventricular dysfunction, patients had a tendency to increase cardiac output (1.06, 95% CI [0.16–2.29], $p = 0.090$), and remarkably reduction in systolic pulmonary artery pressure (5.57, 95% CI [7.60–3.54], $p < 0.001$) and mean pulmonary artery pressure (1.01, 95% CI [1.64–0.37], $p = 0.002$), 24 h after levosimendan infusion (*Hu et al., 2021*).

In addition, mechanical support such as intra-aortic balloon pumping (IABP) improves cardiogenic shock through left ventricular unloading (*Kushner et al., 2009*). Advances in right ventricular assist device (RVAD), including Impella RP and veno-arterial extracorporeal membrane oxygenation (VA-ECMO) are used to manage cardiogenic shock (*Prutkin, Strote & Stout, 2008*; *Anderson et al., 2015*; *Ouweneel et al., 2016*; *Gramegna et al., 2020*). In a retrospective study of five patients with refractory right ventricular failure post-AMI who were failed to recover from cardiogenic shock after treatment with PCI, inotropic and IABP, 80% of the cases immediately received Impella RP support and improved haemodynamics, with an increase in systolic blood pressure (from $91 \pm 17$ to $136 \pm 13$ mmHg) and decrease in central venous pressure (from $16 \pm 2.5$ to $12 \pm 4$ mmHg) (*Gramegna et al., 2020*).

## Limitations and strengths

A previous study involving RVMI mainly focused on RVMI complied with posterior or inferior wall infarction, while our cohort study investigated the difference in clinical characteristics and prognosis in patients with isolated RVMI and LVMI. There are limitations in this study that are worthwhile to be mentioned. First, the retrospective nature of the study means that conclusion remains to be verified in further prospective studies. Second, the SYNTAX score II or Intermountain Risk Score accurately predict mortality in patients with coronary disease (*Hayıroğlu et al., 2018*; *Cınar et al., 2022*). Due to large number of patients lack of this index, we adjusted the "number of vessels with lesion" in the mortality risk estimation instead. Third, despite extensive adjustment of confounders, such as IPTW and PSM, there remain potential confounders did not sufficiently estimate and may bring bias to the results (*i.e.*, infarct size, SYNTAX score/Intermountain Risk Score, work pressure, lifestyle and mental health). Fourth, the population of this study consisted of three hospitals in Hunan province, which limited the generalizability of our findings nationally and globally. Fifth, missing data of BMI and serum potassium was likely to result in bias in our findings, but we have conducted a multiple imputation strategy for missing data to minimized potential bias, Sixth, this study base on a small-scale population, which may reduce the strength and reliability of our hypothesis. Further research with a large-scale population derived nationally is warranted to be performed to address and validate the observation from this hospital.

## CONCLUSION

This study identified the clinical characteristics, treatment strategies, and prognosis in patients with isolated RVMI with respect to those features in patients with isolated LVMI. Due to the physical difference between the left-side heart and right-side heart, the presentation of isolated RVMI has features such as atrioventricular block, hypotension, and cardiogenic shock. Such presentations changed the therapeutic regimen in isolated RVMI patients. Importantly, the risks of all-cause mortality and cardiovascular mortality in isolated RVMI patients were remarkably lower than those in isolated LVMI. This study revealed a better prognosis for isolated RVMI patients compared to isolated LVMI, which indicates the ischemic region could be considered in AMI risk stratification models for better assessment of risk for adverse clinical events.

### Funding

This study was supported by the Natural Science Foundation of China 82170436, 81870171 (to Jingjing Cai), 81770403, 81974054 (to Hong Yuan), the National Key Research and Development Projects 2019YFF0216305 (to Jingjing Cai), and 2016YFC0900802 (to Hong Yuan). The funders had no role in study design, data collection and analysis, decision to publish, or preparation of the manuscript.

## Grant Disclosures

The following grant information was disclosed by the authors:
Natural Science Foundation of China: 82170436, 81870171, 81770403 and 81974054.
National Key Research and Development Projects: 2019YFF0216305 and 2016YFC0900802.

## Competing Interests

The authors declare that they have no competing interests.

## Author Contributions

- Yuanyuan Chen performed the experiments, analyzed the data, prepared figures and/or tables, authored or reviewed drafts of the article, and approved the final draft.
- Chunyan Weng performed the experiments, analyzed the data, prepared figures and/or tables, authored or reviewed drafts of the article, and approved the final draft.
- Junru Wu performed the experiments, prepared figures and/or tables, collected and revised data, and checked the final manuscript, and approved the final draft.
- Yan Tang performed the experiments, prepared figures and/or tables, collected and revised data, and checked the final manuscript, and approved the final draft.
- Zhengxin Li performed the experiments, prepared figures and/or tables, collected and revised data, and checked the final manuscript, and approved the final draft.
- Qing Wen performed the experiments, prepared figures and/or tables, collected and revised data, and checked the final manuscript, and approved the final draft.
- Xuejing Sun performed the experiments, prepared figures and/or tables, collected and revised data, and checked the final manuscript, and approved the final draft.
- Mingxing Wu analyzed the data, authored or reviewed drafts of the article, revised data and the final manuscript, and approved the final draft.
- Zhiliu Peng analyzed the data, authored or reviewed drafts of the article, revised data and the final manuscript, and approved the final draft.
- Xiying Luo analyzed the data, authored or reviewed drafts of the article, revised data and the final manuscript, and approved the final draft.
- Hong Yuan conceived and designed the experiments, authored or reviewed drafts of the article, and approved the final draft.
- Yao Lu conceived and designed the experiments, authored or reviewed drafts of the article, and approved the final draft.
- Wei Huang conceived and designed the experiments, authored or reviewed drafts of the article, and approved the final draft.
- Jingjing Cai conceived and designed the experiments, authored or reviewed drafts of the article, and approved the final draft.

## Human Ethics

The following information was supplied relating to ethical approvals (*i.e.*, approving body and any reference numbers):

The ethics committee of the Third Xiangya Hospital, Central South University approved the study (NO: 2019-S489).

## Data Availability

The raw data are available in the Supplemental Files.

## Supplemental Information

Supplemental information for this article can be found online at http://dx.doi.org/10.7717/peerj.14959#supplemental-information.

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
