# Peer review of "Clinical characteristics and prognosis differences between isolated right and left ventricular myocardial infarction in the Chinese population: a retrospective study"

_PeerJ, doi:10.7717/peerj.14959_

## Round 0.1 · original submission · Major Revisions

Please respond to the reviewers point by point.

Reviewer 2 has requested that you cite specific references. You may add them if you believe they are especially relevant. However, I do not expect you to include these citations, and if you do not include them, this will not influence my decision.

Reviewer 1 ·

Basic reporting

I have reviewed the manuscript entitled 'Clinical characteristics and prognosis differences
between isolated right and left ventricular
myocardial infarction in the chinese population'

The manuscript is well-written however the subject is not a new one in cardiology era.
First the grammar and typo errors should be corrected( what is serum kalium level??)
In addition, following the PCI to the IRA, the situation and critical stenosis in other coronary arteries are very important in the prognosis of these patients. SYNTAX score should be calculated in order to say that isolated RVMI has a better prognosis.
The role of SYNTAX score should be emphasized in the introduction part citing 'Predictive value of SYNTAX score II for clinical outcomes in cardiogenic shock underwent primary percutaneous coronary intervention; a pilot study'.
If the authors did not have the SYNTAX score , it should also be added to the limitation part.

Experimental design

The article is in the standards of this journal.

Validity of the findings

The statistical analyses appear to be very valuable.

Reviewer 2 ·

Basic reporting

The present retrospective study aimed to determine basic clinical and long-term prognosis of patients who were either diagnosed with isolated LVMI and RVMI. The results revealed that even though patients with isolated RVMI had higher incidence of cardiogenic shock and complete AV block at presentation, their long-term prognosis is much better than those with isolated LVMI.

Experimental design

No comment

Validity of the findings

No comment

Additional comments

My remarks about the study:
1- The authors stated that the data was collected from three different stetting. So, is there major difference in terms of patient management among these hospitals?
2- The authors should provide how they chose the best model for multivariable analysis?
3- What do the authors mean by saying serum kalium? Is it calcium?
4- The authors should provide, if applicable, the frequency of no-reflow, contrast-induced nephropathy or new-onset AF in patients with isolated RVMI and LVMI.
5- I am not familiar with Chinese Center for Disease Control and Prevention (CDC) database. How is it reliable for long-term follow-up data?
6-In a recent study, the intermountain risk score is found to be a reliable risk score for the prognosis of ACS patients. Did the authors examine this risk score in their study? If not, please mention and cite it in the limitation section of the article
-Evaluation of Intermountain Risk Score for Short- and Long-Term Mortality in ST Elevation Myocardial Infarction Patients. Angiology. 2022 May 29:33197221105753. doi: 10.1177/00033197221105753. Epub ahead of print. PMID: 35635200.
Thank you.

---

## Round 0.2 · accepted · Accept

All the comments have been addressed properly. I think it could be accepted now.

Reviewer 1 ·

Basic reporting

The authors submitted the manuscript all compatible with the journal

Experimental design

The researh question is well-designed.

Validity of the findings

All the data have been well reported. Conclusions are well stated.

Additional comments

Thank you for the required revisions.

Reviewer 2 ·

Basic reporting

Thank you for revision.
I have no further comments.

Experimental design

I have no further comments.

Validity of the findings

I have no further comments.

Additional comments

Thank you for revision.
I have no further comments.